# Effect of Oxidation Process on Mechanical and Tribological Behaviour of Titanium Grade 5 Alloy

**DOI:** 10.3390/ma17040776

**Published:** 2024-02-06

**Authors:** Abdulsalam Saier, Ismail Esen, Hayrettin Ahlatci, Esma Keskin

**Affiliations:** 1Mechanical Engineering Department, Karabuk University, Karabuk 78050, Turkey; abdulsalamsaier@gmail.com; 2Metallurgical and Materials Engineering Department, Karabuk University, Karabuk 78050, Turkey; hahlatci@karabuk.edu.tr (H.A.); keskinesma5@gmail.com (E.K.)

**Keywords:** Titanium Grade 5, oxidation process, mechanical properties, tribological behaviour, microstructure

## Abstract

In this study, microstructural characterization, mechanical (tensile and compressive) properties, and tribological (wear) properties of Titanium Grade 5 alloy after the oxidation process were examined. While it is observed that the grey contrast coloured α grains are coaxial in the microstructures, it is seen that there are black contrast coloured β grains at the grain boundaries. However, in oxidised Titanium Grade 5, it is possible to observe that the α structure becomes larger, and the number and density of the structure increases. Small-sized structures can be seen inside the growing α particles and on the β particles. These structures are predicted to be Al-Ti/Al-V secondary phases. The nonoxidised alloy matrix and the OL layer exhibited a macrolevel hardness of 335 ± 3.21 HB and 353 ± 1.62 HB, respectively. The heat treatment increased Vickers microhardness by 13% in polished and etched nonoxidised and oxidised alloys, from 309 ± 2.08 HV1 to 352 ± 1.43 HV1. The Vickers microhardness value of the oxidised sample was 528 ± 1.74 HV1, as a 50% increase was noted. According to their tensile properties, oxidised alloys showed a better result compared to nonoxidised alloys. While the peak stress in the oxidised alloy was 1028.40 MPa, in the nonoxidised alloy, this value was 1027.20 MPa. It is seen that the peak stresses of both materials are close to each other, and the result of the oxidised alloy is slightly better. When we look at the breaking strain to characterise the deformation behaviour in the materials, it is 0.084 mm/mm in the oxidised alloy; In the nonoxidised alloy, it is 0.066 mm/mm. When we look at the stress at offset yield of the two alloys, it is 694.56 MPa in the oxidised alloy; it was found to be 674.092 MPa in the nonoxidised alloy. According to their compressive test properties, the maximum compressive strength is 2164.32 MPa in the oxidised alloy; in the nonoxidised alloy, it is 1531.52 MPa. While the yield strength is 972.50 MPa in oxidised Titanium Grade 5, it was found to be 934.16 MPa in nonoxidised Titanium Grade 5. When the compressive deformation oxidised alloy is 100.01%, in the nonoxidised alloy, it is 68.50%. According to their tribological properties, the oxidised alloy provided the least weight loss after 10,000 m and had the best wear resistance. This material’s weight loss and wear coefficient at the end of 10,000 m are 0.127 ± 0.0002 g and (63.45 ± 0.15) × 10^−8^ g/Nm, respectively. The highest weight loss and worst wear resistance have been observed in the nonoxidised alloy. The weight loss and wear coefficients at the end of 10,000 m are 0.140 ± 0.0003 g and (69.75 ± 0.09) × 10^−8^ g/Nm, respectively. The oxidation process has been shown to improve the tribological properties of Titanium Grade 5 alloy.

## 1. Introduction

Titanium constitutes 6% of the earth’s crust. However, due to the difficulty of the production process and technological inadequacy, it was not used as an industrial material until the 1940s. Production was started in the 1940s with the method developed by Wilhelm Kroll and named after him. The commercial applicability of the Kroll method caused the US armed forces to become interested in titanium in 1947. This alloy was initially manufactured in 1952 for use in the engine combustion chamber and the engine connection point on the wings of DC-7-type aircraft [1].

Titanium, in its purest form, is a group of metals that are not very heavy. Its density is 4.51 g/cm^3^, which places it in the middle of the density spectrum between steel and aluminium. Titanium also has a higher melting point than iron, at 1668 °C. Furthermore, titanium and its alloys are highly prized engineering materials nowadays for their exceptional corrosion resistance and strength-to-density ratios [2]. It is possible to increase their strength by heat treatment and deformation. The aerospace, automotive, marine, chemical, and biomedical industries are among the many that are seeing an uptick in the usage of titanium and titanium alloys because of these metals’ exceptional biocompatibility qualities, high specific strength, and resistance to corrosion [3,4]. Ti and its alloys are enhanced by the addition of various elements to enhance their chemical composition, mechanical qualities, and tribological characteristics. As it is known, it is the field of tribology that provides the dry sliding condition where two opposing parts are in contact. Dry sliding conditions are the worst wear conditions in terms of wear rate. Titanium alloys have a dramatic rise in their wear rate when subjected to stress in dry sliding circumstances [5]. In other words, despite their super mechanical properties, their low resistance to plastic shear and poor wear resistance due to their relatively high coefficient of friction are factors that limit their widespread use in load-bearing applications [6,7,8,9,10,11,12]. Several surface treatments are employed to enhance the durability and longevity of titanium and its alloys. These techniques include heat treatment, shot peening [13], anodic plasma electrolytic oxidation [14], laser surface modification [15], and plasma-based ion implantation [16]. Additionally, the use of coatings such as diamond-like carbon films [17], PEO coatings [14], and graphene oxide [18] has been explored to enhance the wear resistance of Ti6Al4V alloy. The most frequently preferred methods in surface treatments applied to titanium are oxidation-based surface treatments. An oxide layer (OL) forms on the surface of titanium and its alloys when exposed to high-temperature environments containing oxygen, while an oxygen diffusion zone (ODZ) forms beneath. The stable oxide layer formed serves as a surface protective layer. Titanium can form various forms of oxides (TiO, TiO_2_, Ti_2_O_3_, Ti_3_O_5_). Among the many possible OL variations of TiO_2_, the most prevalent are rutile and anatase [19]. The layer structures of these oxides are quite complicated. The outermost oxide layer is always oxygen-rich titanium dioxide [11,20,21]. While the relatively thin oxide layer increases corrosion resistance, the oxygen diffused inside increases the surface hardness through the solid solution formation mechanism [7,22,23]. Among titanium alloys, Ti6Al4V alloy, also known as Titanium Grade 5, is one of the most important alloys in the alpha-beta category and is an alloy that greatly affects applications in the military, chemical, biomedical, petrochemical, automotive [24,25], and air/spacecraft industries [26,27]. In their study, Dong and Bell [7] used an Amsler tribometer to compare the wear behaviour of untreated and treated Ti6Al4V alloy in a rolling–sliding motion under boundary lubrication conditions. They discovered that the thermal oxidation (TO) process improved the wear resistance of the alloy significantly. When Güleryüz et al. [28] examined the corrosion and wear properties of Ti6Al4V alloy that had been subjected to a thermal oxidation process, they saw the positive effects of the oxidation process. Biswas et al. [29] observed that the microhardness, wear, and corrosion resistance of the surface improved after the oxidation of Ti6Al4V. García Rueda et al. [30] applied the oxidation process to Ti6Al4V alloy and found that its tribological properties improved.

Recently, the surface modification of Ti6Al4V alloys through oxidation, which enhances mechanical resistance, has demonstrated significant potential. The creation of a thin and mechanically stable oxide layer (OL) supported by an oxide diffusion zone (ODZ) is the key to achieving effective surface protection for titanium and its alloys. This process can be carried out at ambient temperature. There is a limited body of research investigating the thermal oxidation characteristics of Ti6Al4V alloys. Consequently, there is significant interest in studying the oxidation and wear behaviour of titanium and its alloys. Titanium Grade 5 alloy, also known as Ti6Al4V, is the most commonly used titanium alloy, making up almost 50% of global titanium production. While the literature indicates that easy oxidation occurs at a temperature of 600 °C for a duration of 60 h, there is limited research on the combined analysis of tensile and compression properties, the impact on the substrate/matrix, as well as wear behaviours. This study aims to innovate by investigating the mechanical and tribological behaviour of Titanium Grade 5 alloy under specific oxidation conditions.

## 2. Materials and Methods

The chemical composition of Titanium Grade 5 alloy is listed in Table 1. Table 1 shows the X-ray fluorescence (XRF Rigaku ZSX Primus II; Tokyo, Japan) of the alloy by weight. X-ray diffractometry (XRD Rigaku Ultima IV; Tokyo, Japan) at 10–90° and 3°/min was used to determine the phases of the alloy. Analysis of XRD diffraction peaks was performed on an external PC connected to the device, MS Windows^®^ Operating System, and SmartLab Studio-II software 2023.

Even in room temperature air, a very thin OL forms on surfaces coated with titanium because of the metal’s strong affinity for oxygen [24,25]. The OL’s thickness grows substantially when the temperature rises beyond 200 °C [24,25]. The ODZ phase emerges at temperatures greater than 400 °C [24,25]. The production of a thicker but faulty OL, which allows oxygen to easily penetrate the substrate, is caused by an increase in the oxidation rate at temperatures over 600 °C [24,25]. Titanium and its alloys experience a significant decrease in their high-temperature performance when subjected to oxidation at high temperatures, particularly at 800 °C [24,25]. This degradation occurs for two reasons: first, the material becomes thinner as a consequence of the separation of thick OL, and second, a broader ODZ is formed [24,25]. Hence, in the current study, 600 °C and 60 h were selected as the parameters for the oxidation procedure. During the oxidation process, the samples were reached to the given temperature in an oven at normal atmospheric pressure with a heating rate of 10 °C/min. The alloys that had finished their oxidation cycle had cooled in the furnace.

The Titanium Grade 5 sample was cut in the dimensions of 10 × 20 × 10 mm from each of the oxidation and nonoxidation states. A water-cooled band saw was used for the cutting procedure. Once the cutting procedure was finished, the samples were put through an automated sanding and polishing apparatus of the Mikrotest brand. Grit sizes 320, 400, 600, 800, 1000, and 2500 were applied to sandpaper during the sanding process. The polishing processes were finished using a 3 μm Al_2_O_3_ liquid solution after sanding. For the etching process, the etching solution made with 1 mL of HF, 4 mL of HNO_3_, and 5 mL of ethanol were used. An optical microscope (LOM-Carl Zeiss light optical microscope) was used to see the change of grains in phase structure; an electron microscope (SEM-Carl Zeiss Ultra Plus scanning electron microscope, zeiss group Munich, Germany) and EDX (energy dispersion X-ray spectrometry, zeiss group munich, Germany) were investigated to reveal the secondary phases. Brinell macrohardness measurements were conducted on the OL layer. Brinell hardness was determined by subjecting 2.5 mm steel balls to a force of 187.5 N. The Brinell hardness value was evaluated by taking the average of at least 5 measurements. In addition to this macrohardness test, Vickers hardness was performed as a microhardness test with a 135° diamond pyramid tip under a 1 kg load. Vickers hardness value was evaluated by taking the average of at least 5 measurements.

Titanium Grade 5 samples, both in their oxidation and nonoxidation states, were pre-prepared into tensile test specimens with dimensions of 8 mm width and 180 mm length. The Zwick/Roell Z600 tensile apparatus (Zwick/Roell, Munich, Germany) was used for all tensile tests, which were conducted at room temperature with a tensile speed of 1.67 × 10^−3^ s^−1^. Titanium Grade 5 samples, both oxidation and nonoxidation, were created for the compression test with dimensions of 12 mm in diameter and 10 mm in length. A Zwick/Roell Z600 tensile apparatus was used to conduct all compression tests at room temperature with a compression rate of 0.5 mm/min. After the tensile and compression tests of each sample, the fracture surfaces were examined with SEM, and the fracture mechanisms were determined.

A pin-on-disc wear tester (ASTM G99)(Karabuk University, Karabuk, Turkey) [31] was used for wear testing (Figure 1) to compare the wear performance of Titanium Grade 5 alloy, both oxidation and nonoxidation, aiming to simulate mutual flat surface friction conditions. The pin-on-disc wear tester provides contacts that determine the wear behaviour of coated surfaces that are exposed to the atmosphere at a certain humidity and/or in contact with a liquid under different applied loads [26,27]. Wear testing of Titanium Grade 5 alloys in both oxidation and nonoxidation stages was conducted using cylindrical samples of 8 mm in diameter and 20 mm in height. The surfaces to be worn were polished with 0.25 µm diamond suspension for 1–2 min and cleaned with alcohol. The initiation of the wear process in the wear tests occurred at the uppermost OL layer of the oxidised samples. The wear tests were carried out at an average temperature of 24 °C, with a load of 20 N [32,33,34,35,36], a sliding speed of 0.5 m/s, a total sliding distance of 10,000 m, and an ideal humidity level of 45% RH. A 4140-quality steel disc with 59 HRC hardness was used as the counter material determined. The wear results were evaluated by taking the average of the weight losses of at least 3 samples under the given conditions. This 20 N of the load produces a stress of approximately 0.397 MPa. The wear mechanism was examined using scanning electron microscopy (SEM) and energy dispersive X-ray spectroscopy (EDX) to account for variations in alloy element quantity and wear load following the wear test.

## 3. Results and Discussion

### 3.1. XRD Patterns

X-ray diffraction (XRD) analysis is a powerful technique for characterising the phase composition of materials. The dual-phase titanium alloy known as Titanium Grade 5 possesses exceptional comprehensive qualities and combines the best features of the α-phase and β-phase phases [37,38]. A hexagonal (hcp) crystal structure characterises the α phase of Titanium Grade 5, in contrast to the body-centred cubic (bcc) structure seen in the β phase [39]. As seen in the XRD analysis given in Figure 2, the intensity of the α phase and β phase is noteworthy in almost all of the peaks. In the peaks starting at 10°, the first phase was Al_45_V_7_ at 12.5°. In the XRD analysis performed at 10–90°, the highest peak appeared at 38°. In this peak, five different phases were revealed: α-Ti, β-Ti, AlTi, AlTi_3_, and Al_45_V_7_. In the XRD peaks, α-Ti, β-Ti, Al_3_V, and AlTi_3_ phases were last observed at 87°.

The formation of an oxide layer on the surface of the Titanium Grade 5 sample with an oxidation process at 600 °C for 60 h has been proven by the XRD result shown in Figure 2b. After the oxidation process, β-Ti was observed in the structure of the Titanium Grade 5 sample substrate adjacent to the coating/metal interface. In addition, the oxide layer remained thin as the titanium substrate exhibited minimal absorption of oxygen atoms. The predominant components of the oxide coating consist primarily of TiO, Ti_2_O, and Ti_2_O_3_, together with TiO_2_ rutile compounds. Due to the thinness of the oxide layer formed by the contribution of the atmospheric cycle, weak peaks belonging to the AlTi_3_ and β-Ti phases on the XRD patterns were obtained from the matrix.

### 3.2. Microstructure

LOM images of nonoxidised Titanium Grade 5 and oxidised Titanium Grade 5 alloys are shown in Figure 3. While it is observed that the α grains with grey contrast colour in the microstructures are equiaxed, it is observed that β grains with black contrast colour are at the grain boundaries. However, it is possible to observe that in oxidised Titanium Grade 5, the α structure becomes larger, and the number and density of the structure increases. When you look carefully at the photographs regarding this structure, small-sized structures can be seen inside the growing α particles and on the β particles. These structures are predicted to be the Al-Ti/Al-V secondary phases visible in the XRD (Figure 2) analysis.

Figure 4a,b depict SEM micrographs of nonoxidised Titanium Grade 5 and oxidised Titanium Grade 5 alloys. Table 2 demonstrates the EDX analysis of the second phase with distinct morphologies labelled (1–6) in Figure 4a,b. When we look at the EDX analysis that supports the SEM analysis, it can be seen that alpha and beta Ti phases, which are also visible in the XRD analysis (See XRD (Figure 2a)), are present throughout the alloys. In the SEM of nonoxidised Titanium Grade 5 alloy, there is a rod-like structure with a white contrast colour at point 1. Since vanadium is minimal in these structures, these structures are likely Al_3_V. At point 2, sphere–ellipse-like structures are thought to consist of Al-Ti binary intermetallics. At point 3, there is a lamellar structure with white colour on grey contrast, and Al-Ti intermetallics are present here as well. In the SEM of the oxidised Titanium Grade 5 alloy, the presence of vanadium in a light grey contrast structure, which is quite visible and takes up a lot of space, at point 4 attracts attention. The presence of vanadium here is higher than in the nonoxidised alloy, and the structure is thought to consist of Al-V intermetallics. At point 5, the lamellar structure in the oxidation-free alloy has been replaced by a tiny, spherical, white contrast structure, and this structure consists of Al-Ti binary intermetallics. There are also Al-Ti phases in a rectangular-like structure at point 6. Several studies [40,41,42,43,44] have shown the natural occurrence of α_2_-Ti3Al and γ-TiAl intermetallic compounds in Ti-Al alloys that have a high concentration of titanium.

Figure 5 shows the SEM micrograph of the oxide layer on oxidised Titanium Grade 5 alloy. Table 3 shows the EDX analysis of the second stage with different morphologies labelled (1–5) in Figure 5. The thickness of the OL layer formed after oxidation was measured to be approximately 3.33 μm. According to the SEM and EDX analysis results (Table 3), it is assumed that TiO_2_ oxide by weight is formed in region 1, which contains the highest oxygen (Figure 2b). It is seen that both oxygen and titanium values are close to each other in the regions between 2 and 4. In these regions, peaks belonging to TiO and Ti_2_O oxides (Figure 2b), most likely together with TiO_2_, were obtained. In the EDX analysis (Table 3), Ti and a small amount of V, along with oxide content, were observed in the 5th region at the end of the 4th region from the surface to the centre. According to the XRD results given in Figure 2b, it was found that β-Ti and Ti_2_O_3_ peaks appeared in this last region.

### 3.3. Hardness Test Results

Table 4 presents a comparison of the hardness (both macro and micro) obtained from the polished and etched matrix and OL layer of nonoxidised and oxidised Titanium Grade 5 alloys. The nonoxidised alloy matrix yielded a macrolevel hardness value of 335 ± 3.21 HB, while a hardness value of 353 ± 1.62 HB was obtained by measuring the ball trace generated on the OL layer. Since the macrohardness values could not reveal the difference of the heat treatment applied, firstly, the Vickers microhardness taken from the polished and etched matrix of the nonoxidised and oxidised alloy increased by 13%, from 309 ± 2.08 HV1 to 352 ± 1.43 HV1, respectively. This can be attributed to the fact that the β-Ti phase of the matrix (Figure 4) of the Titanium Grade 5 alloy increased slightly after the 600 °C-60 h oxidation process. The Vickers microhardness value obtained from the OL of the oxidised sample was 528 ± 1.74 HV1. It was noted that this value rose by about 50%. Urban et al. [40] and Rastkar et al. [45] have also noticed an enhancement in hardness as a result of the creation of AlTi and AlTi_3_ phases. The hardness of Ti6Al4V alloy was increased with the formation of an oxide layer, independent of the oxidation procedure [46]. The dependence of oxide layer composition and hardness on oxidation process parameters, particularly noting the increased hardness of TiO_2_ oxide layers compared to the substrate [47]. Furthermore, there was observed a substantial increase in hardness for Ti6Al4V alloy following oxidation at 600 °C for 60 h, which has indicated the influence of oxidation conditions on hardness enhancement [48].

### 3.4. Mechanical Test Results

Tensile test results of nonoxidised and oxidised Titanium Grade 5 alloys are given in Figure 6a. As can be seen from the graph, oxidised alloy showed a better result compared to nonoxidised alloy. While the peak stress in the oxidised alloy was 1028.40 MPa, in the nonoxidised alloy, this value was 1027.20 MPa. It is seen that the peak stresses of both materials are close to each other, and the result of the oxidised alloy is slightly better. When we look at the breaking strain to characterise the deformation behaviour in the materials, it is 0.084 mm/mm in the oxidised alloy; in the nonoxidised alloy, it is 0.066 mm/mm. When we look at the stress at offset yield of the two alloys, it is 694.56 MPa in the oxidised alloy; it was found to be 674.09 MPa in the nonoxidised alloy.

Compression test results of nonoxidised and oxidised Titanium Grade 5 alloys are given in Figure 6b. According to Figure 5, the maximum compressive strength is 2164.32 MPa in the oxidised alloy; in the nonoxidised alloy, it is 1531.52 MPa. While the yield strength was 972.50 MPa in oxidised Titanium Grade 5, it was found to be 934.16 MPa in nonoxidised Titanium Grade 5. As can be seen from this graph, when the compressive deformation oxidised alloy is 100.01%, in the nonoxidised alloy, it is 71.56%.

SEM fractured surface images of nonoxidised and oxidised Titanium Grade 5 alloys after the tensile test are given in Figure 7. The nonoxidised alloy is more brittle and has a tougher fracture compared to the oxidised alloy. It is observed that the oxidised alloy causes a more ductile fracture and more deformation when breaking.

SEM fractured surface images of nonoxidised and oxidised Titanium Grade 5 alloys and a compression test are given in Figure 8. In the nonoxidised alloy, in some parts, the part has broken off from the material surface, and these particles are plastered on the matrix surface. It is seen that there are deposits resulting from plastic deformation in the oxidised alloy, and these deposits give a wavy appearance in the broken surface images.

### 3.5. Tribological (Wear) Test Results

The wear weight loss and wear coefficient results of nonoxidised and oxidised Titanium Grade 5 alloys are given in Figure 9 and Figure 10, respectively. The oxidised alloy provided the least weight loss after 10,000 m and had the best wear resistance. This material’s weight loss after a sliding distance of 10,000 m is 0.127 ± 0.0002 g, and its wear coefficient is (63.45 ± 0.15) × 10^−8^ g/Nm. The nonoxidised alloy exhibits the greatest weight loss and the poorest wear resistance. The weight loss and wear coefficient after a sliding distance of 10,000 m are recorded as 0.140 ± 0.0003 g and (69.75 ± 0.09) × 10^−8^ g/Nm, respectively. The oxidation process has been shown to improve the tribological properties of Titanium Grade 5 alloy.

Abrasive and adhesive wear types are observed as the effective wear mechanisms in the SEM micrographs. The sticky wear type is in the form of adhesion of pieces plucking from the surface due to the adhesion of worn parts, and the adhesive wear type is more dominant in the wear mechanism. A noticeable rise in the oxygen peak was noted in the surface EDX study, indicating the presence of an oxidative wear process. The wear micrographs reveal that wear debris, which has been hardened by plastic deformation and oxidation (as indicated in Table 5), causes the development of wear grooves (as shown in Figure 10) on the surface of the tested sample during friction. This occurrence is indicative of abrasive wear. Also, the abrasion shows signs of the three-body abrasion’s efficacy, which is the sliding abrasion mechanism of the abrasive wear debris particle. As seen at point 1 in the nonoxidised alloy (Figure 11a), small particles broken off by the effect of wear are stuck to the matrix surface. At point 2, broken pieces can be seen densely plastered on the matrix surface. At point 3, a deep abrasive wear mark is observed. It can be seen that the wear grooves at point 4 in the oxidised alloy (Figure 11b) are not very deep. Swollen flakes in the form of sticky crusts are observed at point 5 due to adhesive wear. It is thought that the particles separated from the surface due to this flaking support three-body wear. At point 6, it is seen that the broken pieces are stuck inside the shallow wear lines. The formation of the oxide layer on the surface of the oxidised Titanium grade 5 sample leads to a shift in the wear mechanism from abrasive wear to adhesive wear. This finding is attributed to the somewhat greater peak of Ti and the presence of an oxygen peak in Table 5.

## 4. Conclusions

After oxidation, Titanium Grade 5 alloy was studied for its microstructural characteristics and its mechanical (tensile and compression) and tribological (wear) capabilities. The results are as follows:While it is observed that the grey contrast coloured α grains are coaxial in the microstructures, it is seen that there are black contrast coloured β grains at the grain boundaries. However, in oxidised Titanium Grade 5, it is possible to observe that the α structure becomes larger, and the number and density of the structure increases. Small-sized structures can be seen inside the growing α particles and on the β particles. These structures are predicted to be Al-Ti/Al-V secondary phases.The nonoxidised alloy matrix had a macrolevel hardness of 335 ± 3.21 HB, while the ball trace on the OL layer showed a hardness of 353 ± 1.62 HB. The heat treatment used resulted in a 13% increase in Vickers microhardness from the polished and etched matrix of the nonoxidised and oxidised alloys, from 309 ± 2.08 HV1 to 352 ± 1.43 HV1, respectively. The Vickers microhardness value of the oxidised sample was 528 ± 1.74 HV1, as a 50% increase was noted.According to their tensile properties, oxidised alloys showed a better result compared to nonoxidised alloys. While the peak stress in the oxidised alloy was 1028.40 MPa, in the nonoxidised alloy, this value was 1027.20 MPa. It is seen that the peak stresses of both materials are close to each other, and the result of the oxidised alloy is slightly better. When we look at the breaking strain to characterise the deformation behaviour in the materials, it is 0.084 mm/mm in the oxidised alloy; in the nonoxidised alloy, it is 0.066 mm/mm. When we look at the stress at offset yield of the two alloys, it is 694.56 MPa in the oxidised alloy; it was found to be 674.092 MPa in the nonoxidised alloy.According to their compressive test properties, the maximum compressive strength is 2164.32 MPa in the oxidised alloy; in the nonoxidised alloy, it is 1531.52 MPa. While the yield strength is 972.50 MPa in oxidised Titanium Grade 5, it is found to be 934.16 MPa in nonoxidised Titanium Grade 5. When the compressive deformation oxidised alloy is 100.01%, in the nonoxidised alloy, it is 68.50%.According to their tribological properties, the oxidised alloy provides the least weight loss after 10,000 m and has the best wear resistance. The oxidised alloy experiences a weight loss of 0.127 ± 0.0002 g after sliding a distance of 10,000 m. Additionally, its wear coefficient is (63.45 ± 0.15) × 10^−8^ g/Nm. The unoxidised alloy exhibits the highest degree of weight reduction and the lowest level of wear resistance. The weight loss and wear coefficient, measured after a sliding distance of 10,000 m, is determined as 0.140 ± 0.0003 g and (69.75 ± 0.09) × 10^−8^ g/Nm, respectively. The oxidation process is shown to improve the tribological properties of Titanium Grade 5 alloy.

## Figures and Tables

**Figure 1 materials-17-00776-f001:**
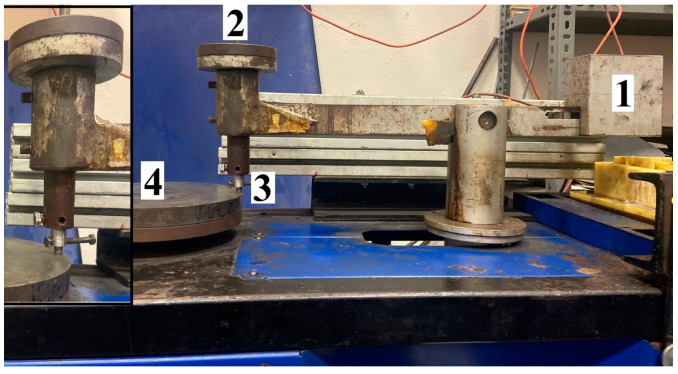
Schematic views of wear testers utilised in this study. 1: counterweight; 2: applied load of 20 N; 3: pin wear test sample; 4: 4140-quality steel disc.

**Figure 2 materials-17-00776-f002:**
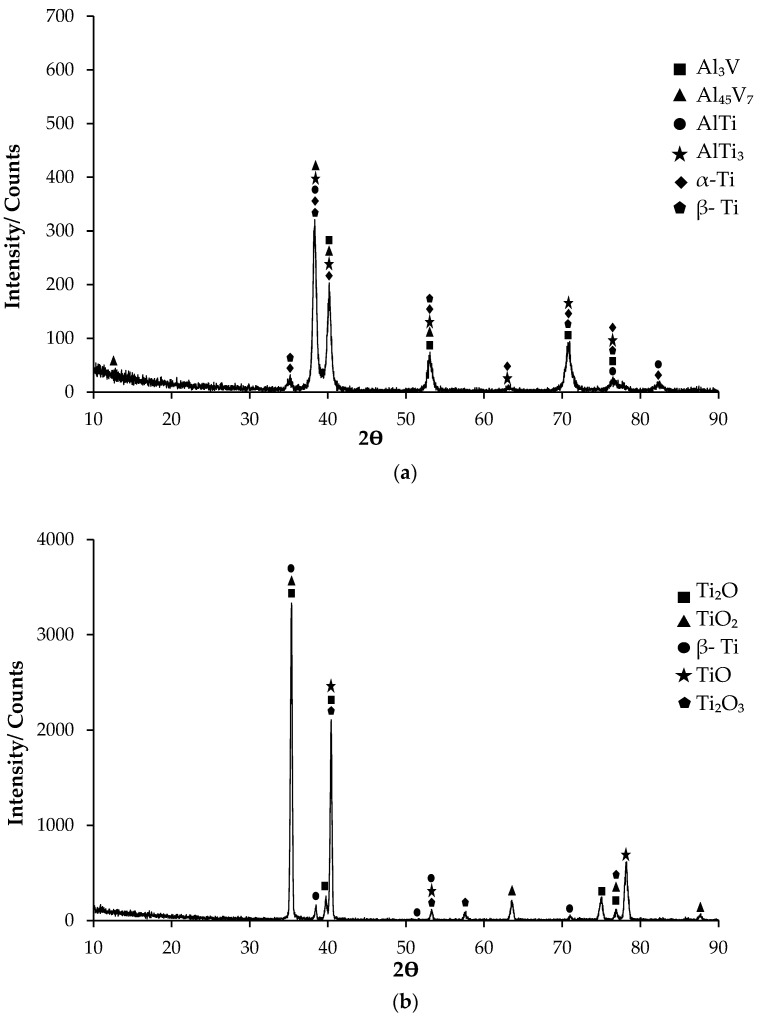
X-ray diffraction (XRD) patterns of Titanium Grade 5 sample: (**a**) Nonoxidised and (**b**) oxidised.

**Figure 3 materials-17-00776-f003:**
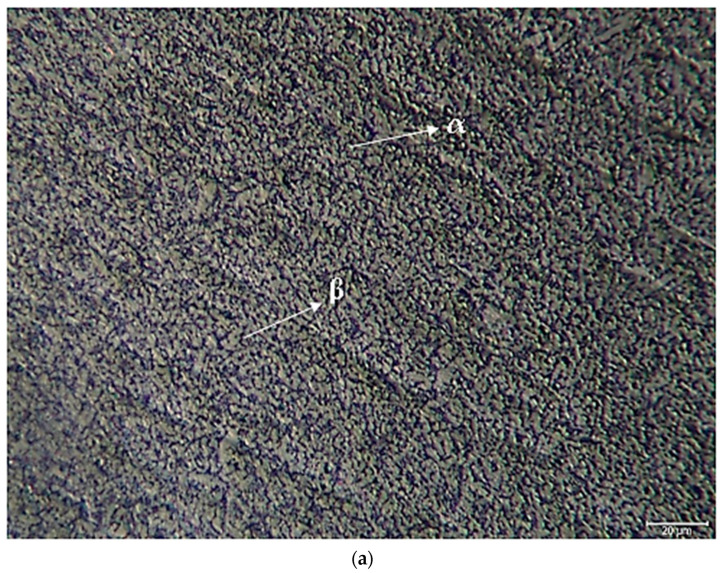
Image of Titanium Grade 5 alloy LOM: (**a**) nonoxidised and (**b**) oxidised.

**Figure 4 materials-17-00776-f004:**
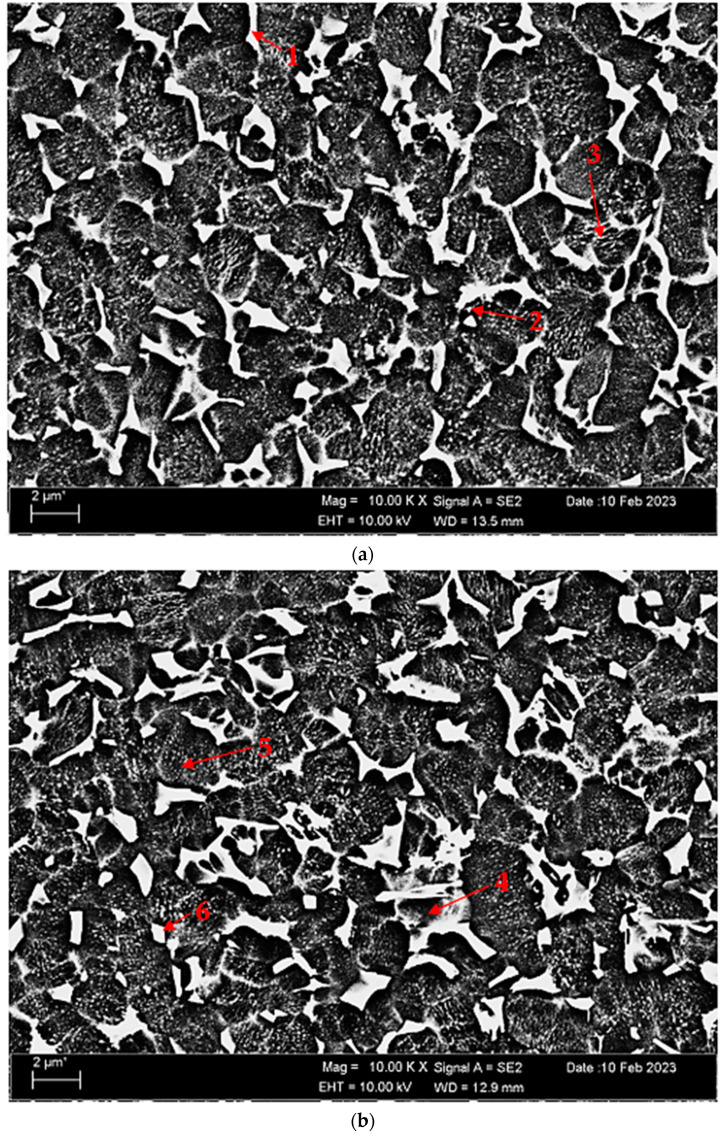
Image of Titanium Grade 5 alloy SEM at 10 kX: (**a**) nonoxidised and (**b**) oxidised.

**Figure 5 materials-17-00776-f005:**
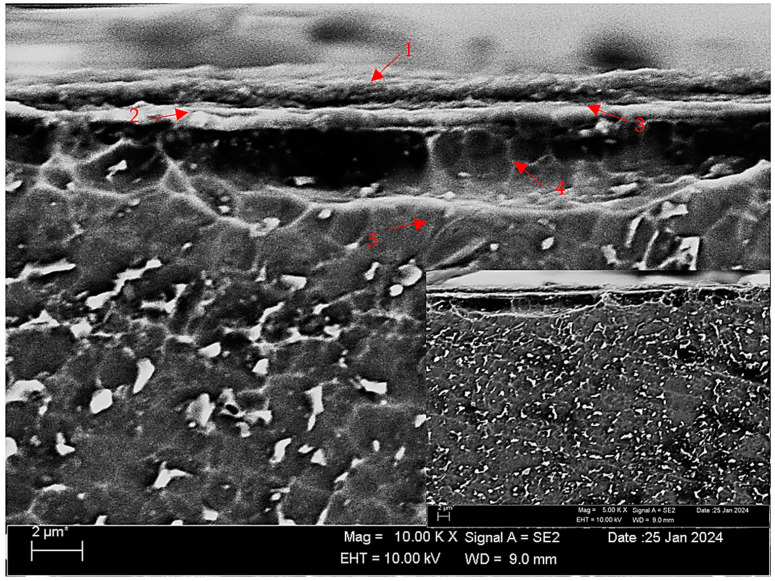
SEM image of oxidised Titanium Grade 5 oxide layer at 10 kX.

**Figure 6 materials-17-00776-f006:**
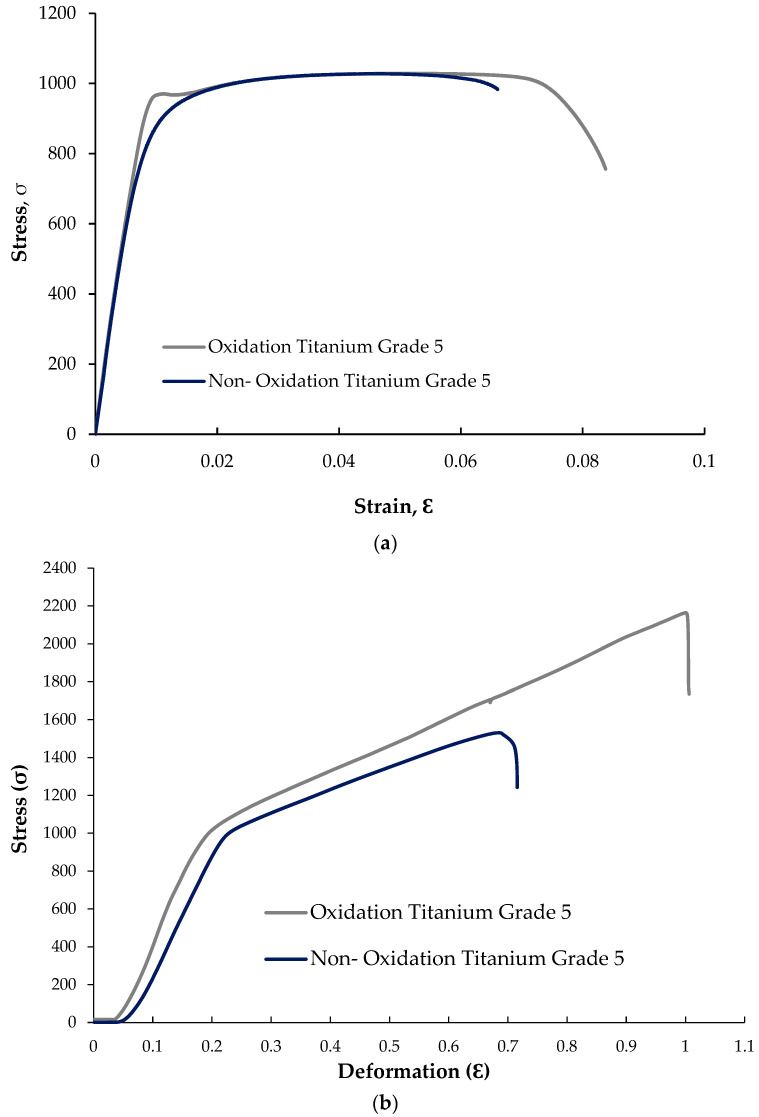
(**a**) Tensile and (**b**) compression test results of Titanium Grade 5 alloy.

**Figure 7 materials-17-00776-f007:**
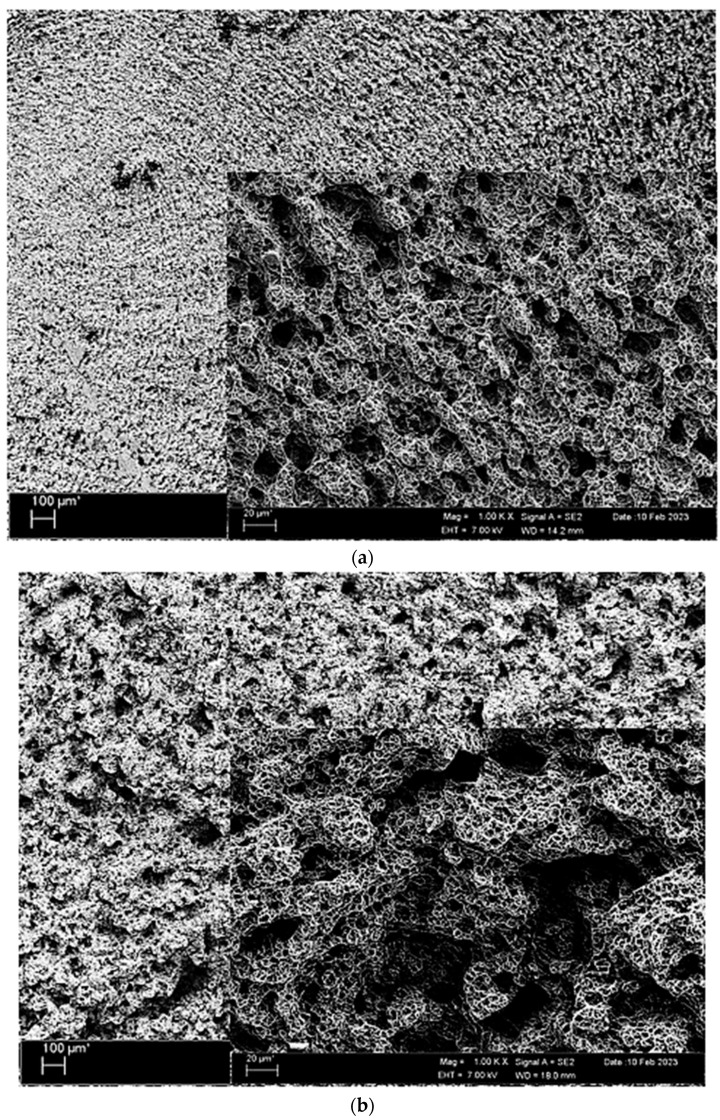
Image of Titanium Grade 5 alloy SEM undergoing tensile: (**a**) nonoxidised and (**b**) oxidised.

**Figure 8 materials-17-00776-f008:**
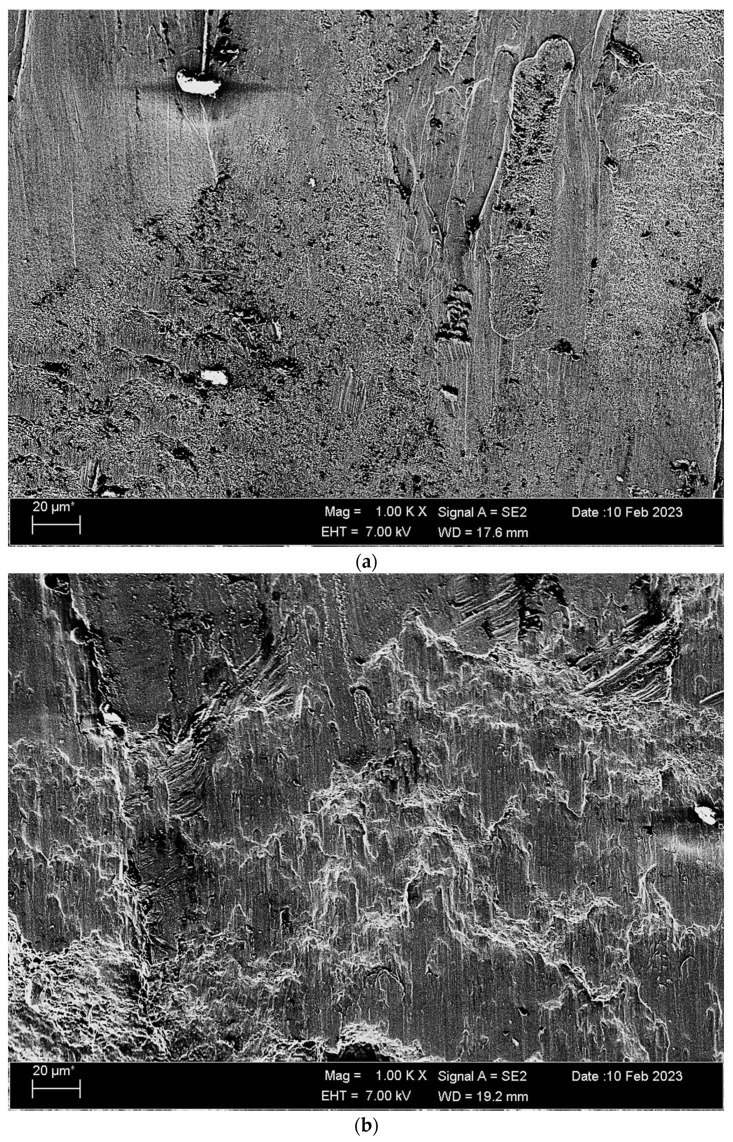
Image of Titanium Grade 5 alloy SEM undergoing compression, at 1 kX: (**a**) nonoxidised and (**b**) oxidised.

**Figure 9 materials-17-00776-f009:**
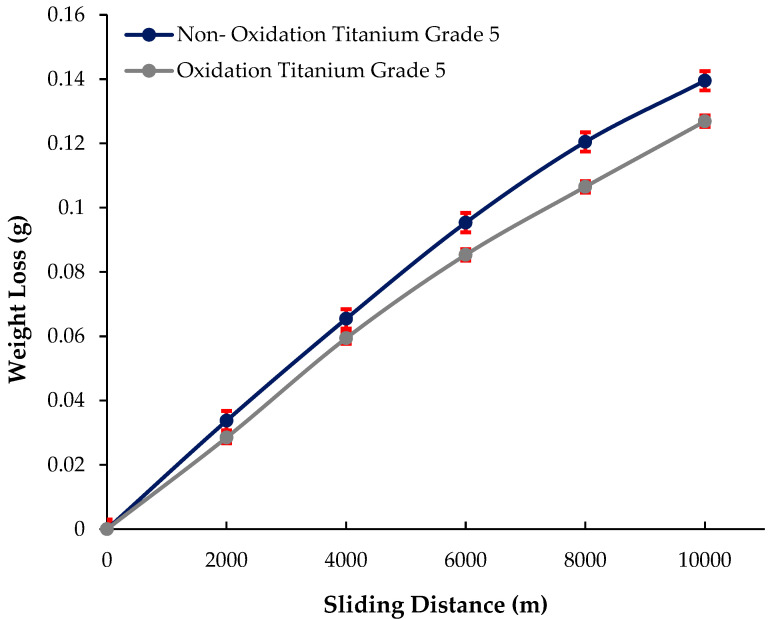
Wear weight loss results of nonoxidised and oxidised Titanium Grade 5 alloys.

**Figure 10 materials-17-00776-f010:**
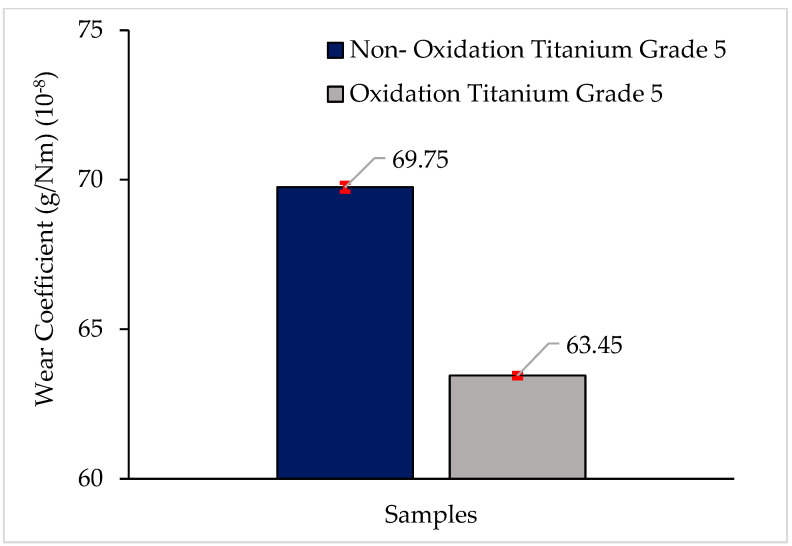
Wear coefficient results of nonoxidised and oxidised Titanium Grade 5 alloys.

**Figure 11 materials-17-00776-f011:**
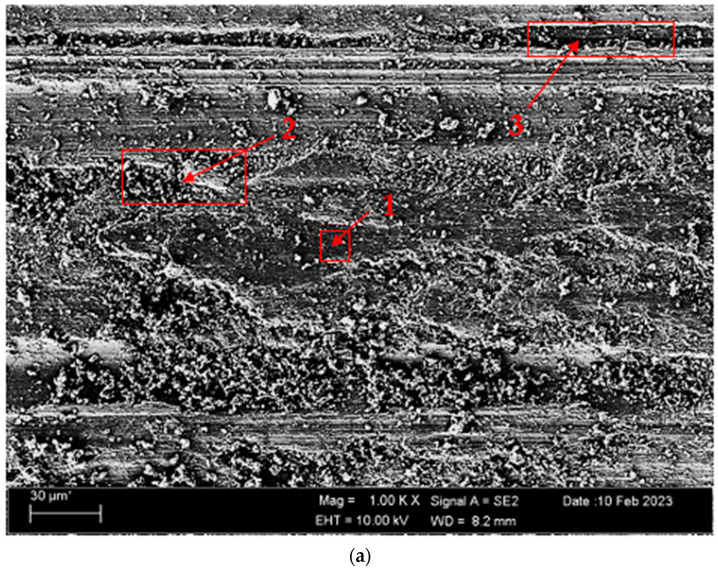
Image of Titanium Grade 5 alloy SEM undergoing wear: (**a**) nonoxidised and (**b**) oxidised.

**Table 1 materials-17-00776-t001:** Alloy compositions (wt%).

	Chemical Composition (wt.%)		
Alloy	Al	V	Ti
Titanium Grade 5	7.07	3.51	Bal.

**Table 2 materials-17-00776-t002:** EDX findings of Figure 4a,b (wt.%).

Points	Al	Ti	V
1	7.69	92.02	0.29
2	8.10	91.91	-
3	8.64	91.36	-
4	7.53	90.45	2.03
5	7.59	92.41	-
6	8.66	91.35	-

**Table 3 materials-17-00776-t003:** EDX findings of Figure 5 (wt.%).

Points	Al	Ti	V	O
1	2.16	41.69	-	56.16
2	0.20	98.27	-	1.53
3	0.31	98.37	-	1.31
4	0.51	97.82	-	1.67
5	7.65	91.38	0.49	0.47

**Table 4 materials-17-00776-t004:** Hardness results of Titanium Grade 5 alloys nonoxidised and oxidised.

Alloy	Hardness Test (Macro and Micro)		
Titanium Grade 5	Nonoxidised	Oxidised	
	Polished and Etched Matrix	Polished and Etched Matrix	OL
Macro (HB/2.5/187.5)	335 ± 3.21	-	353 ± 1.62
Micro (HV1)	309 ± 2.08	352 ± 1.43	528 ± 1.74

**Table 5 materials-17-00776-t005:** EDX findings of Figure 11a,b (wt.%).

Points	Al	Ti	V	Fe	O
1	4.14	53.22	0.18	10.12	32.34
2	3.97	64.40	-	8.23	23.41
3	4.34	77.46	0.10	5.73	12.44
4	4.33	61.50	0.11	9.16	24.89
5	3.50	70.87	1.66	4.53	19.44
6	4.32	61.59	0.44	10.50	23.44

## Data Availability

Data are contained within the article.

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
