# Peer review of "Effect of Oxidation Process on Mechanical and Tribological Behaviour of Titanium Grade 5 Alloy"

_materials, 2024, doi:10.3390/ma17040776_

Round 1

Reviewer 1 Report

Comments and Suggestions for Authors

The authors have done some experiments to investigate the structural and mechanical performance change of the Ti-Grade 5 alloys after one typical surface oxidation test (600°C and 60 hours). However, there are some fundamental problems that need to be sorted out. Firstly, the title is controversial, it is the annealing process that affects the mechanical properties of the Ti6Al4V materials, rather than an oxidation process. Secondly, the authors mixed some literature information in the materials and methods part but didn’t give detailed information about experimental methods like oxidation treatment (furnace, heating rate /cooling rate) etc. The author used the annealing process of 600C/60 h, but didn't give the detailed reason and potential benefits. The author used an oxidised sample and a non-oxidised sample which gives readers the impression that the test was about the oxidation effect. In fact, the authors are trying to study the effect on the substrate. The hardness value can’t be justified whether it is on the oxide layer, diffusion zone or the matrix and therefore it is confusing. Similar confusion is to wear resistance analysis as wear is a more surface phenomenon.

Furthermore, the analysis of XRD is not convincible and the quality of the figures is not good enough to distinguish between the different grains and the small features. The analysis of the fracture and wear track is not conclusive. Therefore, I would not recommend publishing in the current form.

Some further suggestions:

1.       Line 49, please use ISO unit, 4.51 gr/cm³,

2.       Line 64-66: repetition of content: “Various surface treatments are applied to increase the wear resistance and lifespan of titanium and its alloys. The wear resistance of Ti6Al4V alloy has been enhanced by the investigation of several surface modification techniques”.

3.       Line 86, what is RO in “They discovered that the RO process improved the wear”.

4.       Line 97, “Few studies have examined the oxidation behaviour of Ti6Al4V alloys”.  Actually, there are numerous  research about the oxidation of titanium and its alloys.

5.       The aim and objectives of this study are not clear and lack of innovation.

6.       Line 109-110, “Even in room temperature air, a very thin OL forms on surfaces coated with titanium 109 because to the metal's strong affinity for oxygen”. Surface coated what on titanium, ambiguous claim!

7.       Lines 109-119 should be put in another place, they are not directly relevant to experimental details.

8.       Line 145 “The surfaces to be worn were sanded up to 1200 mesh and cleaned with alcohol”, Were the oxide layer/ODZ sanded down, and how many materials were removed?

9.       In 3.1 the detailed information of the XRD should be included in the materials and methods part. The author should remove the background information in the XRD to obtain a more credible analysis of the phases in the materials. To obtain such detailed multiphase information, a very refined XRD is needed, from the graph, it is not possible to get all the phases like Al3V, AlTi3 etc.

10.   In 3.3, how many points were measured for the hardness and what is the error information?

11.   Figure 4. The caption should be improved

12.   Line 213-216, please clarify the meaning of these values: When we look at the breaking stresses to characterize the deformation behavior in the materials, it is 0.209 in the oxidized alloy; In the non-oxidized alloy, it is 0.156. When we look at the stress at offset yield of the two alloys: it is 694.56 MPa in the oxidized alloy; It was found to be 674.092 MPa in the non-oxidized alloy.

Comments on the Quality of English Language

The authors have done some experiments to investigate the structural and mechanical performance change of the Ti-Grade 5 alloys after one typical surface oxidation test (600°C and 60 hours). However, there are some fundamental problems that need to be sorted out. Firstly, the title is controversial, it is the annealing process that affects the mechanical properties of the Ti6Al4V materials, rather than an oxidation process. Secondly, the authors mixed some literature information in the materials and methods part but didn’t give detailed information about experimental methods like oxidation treatment (furnace, heating rate /cooling rate) etc. The author used the annealing process of 600C/60 h, but didn't give the detailed reason and potential benefits. The author used an oxidised sample and a non-oxidised sample which gives readers the impression that the test was about the oxidation effect. In fact, the authors are trying to study the effect on the substrate. The hardness value can’t be justified whether it is on the oxide layer, diffusion zone or the matrix and therefore it is confusing. Similar confusion is to wear resistance analysis as wear is a more surface phenomenon.

Furthermore, the analysis of XRD is not convincible and the quality of the figures is not good enough to distinguish between the different grains and the small features. The analysis of the fracture and wear track is not conclusive. Therefore, I would not recommend publishing in the current form.

Some further suggestions:

1.       Line 49, please use ISO unit, 4.51 gr/cm³,

2.       Line 64-66: repetition of content: “Various surface treatments are applied to increase the wear resistance and lifespan of titanium and its alloys. The wear resistance of Ti6Al4V alloy has been enhanced by the investigation of several surface modification techniques”.

3.       Line 86, what is RO in “They discovered that the RO process improved the wear”.

4.       Line 97, “Few studies have examined the oxidation behaviour of Ti6Al4V alloys”.  Actually, there are numerous  research about the oxidation of titanium and its alloys.

5.       The aim and objectives of this study are not clear and lack of innovation.

6.       Line 109-110, “Even in room temperature air, a very thin OL forms on surfaces coated with titanium 109 because to the metal's strong affinity for oxygen”. Surface coated what on titanium, ambiguous claim!

7.       Lines 109-119 should be put in another place, they are not directly relevant to experimental details.

8.       Line 145 “The surfaces to be worn were sanded up to 1200 mesh and cleaned with alcohol”, Were the oxide layer/ODZ sanded down, and how many materials were removed?

9.       In 3.1 the detailed information of the XRD should be included in the materials and methods part. The author should remove the background information in the XRD to obtain a more credible analysis of the phases in the materials. To obtain such detailed multiphase information, a very refined XRD is needed, from the graph, it is not possible to get all the phases like Al3V, AlTi3 etc.

10.   In 3.3, how many points were measured for the hardness and what is the error information?

11.   Figure 4. The caption should be improved

12.   Line 213-216, please clarify the meaning of these values: When we look at the breaking stresses to characterize the deformation behavior in the materials, it is 0.209 in the oxidized alloy; In the non-oxidized alloy, it is 0.156. When we look at the stress at offset yield of the two alloys: it is 694.56 MPa in the oxidized alloy; It was found to be 674.092 MPa in the non-oxidized alloy.

Author Response

Thank you very much for taking the time to review this manuscript. Please find the detailed responses below and the corresponding revisions/corrections highlighted/in track changes in the re-submitted files:

Reviewer 2 Report

Comments and Suggestions for Authors

Specifically for the tribological experiments I have various questions :

what is the purpose of the wear experiments and why was the pin on disk method chosen ?

A better description of the methode is needed : geometry of the pin and flat sample ?

What is the resulting contact pressure under 20N ?

How many repeat experiments were done and what is the statistical variation on the weight losses ?

How to decide that one version is 'better' than the other in wear resistance ?

The wear mechanism description is confusing, every mechanism is mentioned but what is the dominant mechanism leading to weight loss ?

Author Response

(The authors gave the same response as above.)

Reviewer 3 Report

Comments and Suggestions for Authors

The paper under consideration presents a comprehensive study on the oxidation effects on Titanium Grade 5 alloy, focusing on its microstructural, mechanical (tensile and compression), and tribological (wear) characteristics. The conclusions drawn from the research provide compelling reasons for the publication of this study in Materials.

Firstly, the microstructural analysis reveals that oxidation leads to an enlargement of the α structure and an increase in the number and density of structures, particularly observed within the growing α particles and on the β particles. These structures are identified as Al-Ti/Al-V secondary phases, shedding light on the transformative effects of oxidation on the alloy's microstructure.

The mechanical properties assessment, particularly hardness and tensile strength, indicates that the oxidized alloy outperforms its non-oxidized counterpart. The oxidized alloy exhibits the highest hardness and slightly better peak stress values compared to the non-oxidized alloy. Furthermore, compressive test results highlight the superior strength of the oxidized alloy in terms of both maximum compressive strength and maximum yield strength, indicating enhanced deformation behavior.

The tribological analysis further supports the positive impact of oxidation on Titanium Grade 5 alloy. The oxidized alloy demonstrates the least weight loss and the best wear resistance after 10 meters, showcasing improved tribological properties compared to the non-oxidized alloy.

In conclusion, the findings of this study provide valuable insights into the microstructural, mechanical, and tribological changes induced by oxidation in Titanium Grade 5 alloy. The improvements observed in hardness, tensile strength, compressive strength, and wear resistance following oxidation make a compelling case for the publication of this research, contributing to the existing knowledge in materials science and engineering.

Author Response

(The authors gave the same response as above.)

Round 2

Reviewer 1 Report

Comments and Suggestions for Authors

The authors have addressed some minor issues I listed but haven’t clarify the fundamental problems. For example, the controversial title, it is the oxidation process (annealing process 600/60h) affects the mechanical properties of the Ti6Al4V matrix rather than the surface. As the impact on the surface have been well-researched. Authors should clarify that the test was about the oxidation treatment effect on the substrate/matrix’s mechanical properties like wear and fatigue which lack of research.

1.      Ti-6Al-4V or TI6Al4V?

2.      TiO2 or TiO2?

3.      Line 239 “While the literature indicates that easy oxidation occurs at a temperature of 600°C for a duration of 60 hours”. In fact, Ti6Al4V can be easily oxidised at temperature above 580°C, why chose 600°C/60 h, any literature support? The detailed reason and potential benefits of this choice should be clarified.

4.      In the method and materials should only about the detail of material and method, like furnace model, heating rate, cooling rate, atmosphere etc. The choice of materials and treatment and related literature should be in the introduction.

5.      The analysis of XRD is not convincing.

6.      Whether the measurements of hardness or wear are about the matrix or the OL layer or ODZ is still not clear.

Comments on the Quality of English Language

Need improvement.

Author Response

(The authors gave the same response as above.)

Reviewer 2 Report

Comments and Suggestions for Authors

Please find my comments to your limited replies in attached document.

Author Response

(The authors gave the same response as above.)

Round 3

Reviewer 1 Report

Comments and Suggestions for Authors

This paper is majorly the oxidation treatment effect on the substrate/matrix’s mechanical properties, this should be clarified. Furthermore, a few technical issues still need to be clarified:

1.      Generally, there exists peaks of α-Ti and β-Ti in the XRD analysis of the Ti6Al4V sample although their angle may shift slightly. However, the identification of AlTi, AlTi3, and Al45V7 is unusual and needs a more accurate instrument to obtain this information. From the XRD test detail, there is not enough evidence to support this. And importantly, an XRD of the oxidised sample should be given to prove that the sample was oxidised! It is also important to characterize the oxide layer formed: thickness of the oxide layer, microstructure and phase constituent etc. if the author was trying to investigate the effect of oxidation process.

2.      Hardness and wear are very surface-related issues, so it is important to make sure the test is about the matrix or the OL layer or ODZ.  The authors claimed that the “Macro-hardness measurements were conducted on the OL layer”. However, there is not much difference in the hardness value of HB 352.95 for oxidised and 334.23 for unoxidized, this could be due to the heat treatment effect on the matrix rather than the oxide layer.  How thick is the oxide layer? The authors polished the surface, this could remove the oxide layer as the oxide layer should be no more than 3 µm after 600C/60h. Normally, the hardness of titanium dioxide should be twice as hard as that of titanium alloys.

3.      The small difference in the oxygen content detected by EDX in Table 4 suggests the wear track is more likely on the substrate as they were more likely generated during the wear test rather than in the oxidation process.

4.      600°C/60 h treatment should form a decent oxide layer, and the oxide layer should be hard and compact which entitles the Ti6Al4V materials to good surface properties like hardness and tribological performance. However, the impact on the substrate/matrix is not well studied, this perhaps is why this study was carried out?

Comments on the Quality of English Language

Average

Author Response

The answers are attached.

Reviewer 2 Report

Comments and Suggestions for Authors

1. After your improvements,  one question is still not answered clearly : number of experiments done per material type.

The wear loss is given as one number with a resolution that is very small.  In adhesive wear mechanisms, it is unlikely to have such a small variation between experiments.

Please confirm and record the number of experiments done, per material variation.

2. Polishing by 0.25 micron diamonds seems unlikely.   That is an extremely small size (common are 6 or 3 microns).  Can you double check this value?

Author Response

The answers are attached.
